# Tyrosinase Inhibition Antioxidant Effect and Cytotoxicity Studies of the Extracts of *Cudrania tricuspidata* Fruit Standardized in Chlorogenic Acid

**DOI:** 10.3390/molecules24183266

**Published:** 2019-09-07

**Authors:** Ha-Na Oh, Dae-Hun Park, Ji-Yeon Park, Seung-Yub Song, Sung-Ho Lee, Goo Yoon, Hong-Seop Moon, Deuk-Sil Oh, Sang-Hoon Rhee, Eun-Ok Im, In-Soo Yoon, Jung-Hyun Shim, Seung-Sik Cho

**Affiliations:** 1College of Pharmacy, Mokpo National University, Muan-gun, Jeonnam 58554, Korea; 2Department of Nursing, Dongshin University, Naju-si, Jeonnam 58245, Korea; 3Jeollanam-do Forest Resource Research Institute, Naju, Jeonnam 58213, Korea; 4Department of Biological Sciences, Oakland University, Rochester, MI 48309, USA; 5Department of Pharmacy, College of Pharmacy, Pusan National University, Busan 46241, Korea

**Keywords:** *C.tricuspidata* Bureau, HPLC, tyrosinase

## Abstract

In the present study, various extracts of *C. tricuspidata* fruit were prepared with varying ethanol contents and evaluated for their biomarker and biological properties. The 80% ethanolic extract showed the best tyrosinase inhibitory activity, while the 100% ethanolic extract showed the best total phenolics and flavonoids contents. The HPLC method was applied to analyze the chlorogenic acid in *C. tricuspidata* fruit extracts. The results suggest that the observed antioxidant and tyrosinase inhibitory activity of *C. tricuspidata* fruit extract could partially be attributed to the presence of marker compounds in the extract. In this study, we present an analytical method for standardization and optimization of *C. tricuspidata* fruit preparations. Further investigations are warranted to confirm the in vivo pharmacological activity of *C. tricuspidata* fruit extract and its active constituents and assess the safe use of the plant for the potential development of the extract as a skin depigmentation agent.

## 1. Introduction

*Cudrania tricuspidata* (Moraceae) is used as traditional medicine for inflammation, gastritis, cancer, and liver injury [1]. In the previous reports, active constituents from roots and leaves of *Cudrania tricuspidata* contain pharmaceutically active substances such as neuroprotective [2], anti-inflammatory [3,4], pancreatic lipase inhibitory [5], monoamine oxidase inhibitory [6], and anti-obesity effects [7]. Additively, prenylated isoflavonoids, benzylated flavonoids, xanthones from the fruits displayed potential antioxidant, anti-inflammatory, and neuroprotective activities [8,9,10].

The efficacy of extracts and purified bioactive substances prepared using *C. tricuspidata* as a medical source has been studied broadly to date. The content of a single compound present in fruits was insufficient for use as biomarkers for pharmaceutical/cosmetic application. Moreover, preparations involving the fruit could be beneficial for productivity purpose as *C. tricuspidata* is a perennial plant (Table 1).

Few studies have been conducted on the fruits of *C. tricuspidata* and the contents of bioactive substances were observed to be insufficient for use as key compounds for pharmaceutical industrialization. Considerable effort has been focused on developing *C. tricuspidata* as materials, but no positive results have been achieved.

The aim of this study was to evaluate the fruit extract of *C. tricuspudata* for tyrosinase inhibitory activity, as well as to characterize the chromatographic profile of its optimized extract to identify the compounds responsible for antioxidant and tyrosinase inhibition. Validation of a High Performance Liquid Chromatography (HPLC) method was preformed for standardize of chlorogenic acid.

In the preliminary study, we purified and identified the main substance, chlorogenic acidwith antioxidant and tyrosinase inhibitory activity from fruits of *C. tricuspidata*. Previous reports have demonstrated that chlorogenic acid plays important roles in melanogenesis of B16 melanoma cells. Although chlorogenic acid did not exhibit strong tyrosinase inhibitory effect, its metabolic product(s) showed suppression of melanogenesis in B16 melanoma cells by inhibiting tyrosinase activity [13]. Consequently, we set chlorogenic acid as a biomarker for the extract of *C. tricuspidata* fruit.

Cytotoxicity test was assessed in cell lines to test the cell viability in the presence of the extract of *C. tricuspudata* fruit with an aim to incorporate the extract in topical form as a skin whitening agent. This is the first study that assess tyrosinase inhibition and quantifythe presence of biomarkers such as chlorogenic acid in *C. tricuspudata* fruit.

Previously, we had investigated the biological properties of extracts and their biomarkers obtained from *C. tricuspidata* leaves for the development of medicinal/food sources. In this study, fruit components of *C. tricuspidata* were screened for cosmetic application. Extracts of *C. tricuspidata* fruit were prepared for the assessment of chemical composition and biological properties.

## 2. Results and Discussion

### 2.1. Chromatographic Conditions for Extract of C. tricuspidata Fruit

The HPLC conditions were established as follows. A gradient program was used to separate the chlorogenic acid (Table 6). Detection wavelengths were set as 330 nm. As shown in Figure 1, chlorogenic acid was identified as the main component in the extract from *C. tricuspidata*.

Lee et al. reported that the extraction yield of water extract of *C. tricuspidata* fruit was 12.7% and extract contained rutin [3]. However, the content of rutin in the water extract was not described. In the present study, rutin was not found in the extract of *C. tricuspidata* fruit.

Jiang et al. purified and identified anticancer compound named scandenolone from *C. tricuspidata* fruit [12]. Jiang described the detailed purification process in the reported study. However, the study lacked a description of the content of active compound in the fruits of *C. tricuspidata*. Although it has been reported that scandenolone plays an important role in mediating anticancer activity, the potential of the compound to prevent cancer cannot be guaranteed by just consuming *C. tricuspidata* fruits. In addition, there exists no data on permissible levels of consumption for human. Therefore, scandenolone can be considered as one of the trace components of fruits of *C. tricuspidata*.

Jo et al. reported about anti-obesity efficacy of 6,8-Diprenylgenisteinusing 70% ethanol extract of *C. tricuspidata* fruit in their study [7]. The daily intake was set as 10–15 g of fruit. In the present study, 6,8-Diprenylgenistein was analyzed using HPLC, but it was difficult to confirm its presence in the extract of *C. tricuspidata* fruit. As the species, harvesting time of *C. tricuspidata* fruit, and the places of cultivation are different, we presumed that the presence of 6,8-Diprenylgenisteinmight also be different.

### 2.2. Method Validation

#### 2.2.1. Linearity, Limit of Detection (LOD), and Limit of Quantification (LOQ)

In the present study, calibration curves, limit of detection, and quantification were conducted. Calibration curves were set in the range of 3.125–50 μg/mL for chlorogenic acid and exhibited good linear regressions (*r*^2^ = 0.998). The LOD was found to be 0.7 μg/mL for chlorogenic acid. The LOQ value for chlorogenic acid was found to be 2.1 μg/mL (Table 2).

#### 2.2.2. Precision and Accuracy

The results of the intraday and interday precision experiments are shown in Table 3. The overall recovery percentages were in the range of 104.04–107.78% for chlorogenic acid. These results demonstrate that the developed method is reproducible with a good accuracy (Table 3).

#### 2.2.3. Repeatability

The results of the repeatability are shown in Table 4. RSD values were below 2.0%. Thus, HPLC method is suitable for analysis of *C. tricuspidata* fruit.

### 2.3. Contents of Marker Compounds from C. tricuspidata Fruit Extracts

Plant samples were extracted with various solvent compositions to select the best extraction solvent conditions: hot water, 20–100% ethanol (*v*/*v*). The validated HPLC method was applied to analyze the samples. The contents (%wt.) of chlorogenic acid is presented in Figure 2. The contents of the chlorogenic acid in the 80% ethanolic extract were greater compared to other ethanolic extracts. Based on these results, the80% ethanol was selected as the most effective extraction solvent (0.34 ± 0.01%, *w*/*w*).

### 2.4. Cell Viability and Tyrosinase Inhibition of C. tricuspidata Fruit Extracts

Cytotoxicity of various extracts of *C. tricuspidata* was determined by MTT (3-(4,5-dimethylthiazol-2-yl)-2,5-diphenyltetrazolium bromide) assay [14]. Cytotoxicity was assessed after treatment of B16F10 cells with a various sample concentration of 100 μg/mL for 24 h. For further study, 100 μg/mL or less should be considered as optimal for conducting experiments on unraveling the mechanism of action of *C. tricuspidata* extract (Figure 3).

The tyrosinase inhibition of various extracts of *C. tricuspidata* was determined by the tyrosinase inhibitory assays. The measured tyrosinase inhibitory activity is shown in Figure 4. The tyrosinase inhibition decreased in the following order: 80% ethanol extract (68.3 ± 7.3%) > 100% ethanol extract (64.1 ± 5.2%) > 60% ethanol extract (51.2 ± 1.3%) > 40% ethanol extract (24.5 ± 6.8%) > 20% ethanol extract (10.68 ± 0.4%) > hot water extract (5.22 ± 0.5%). 

In the present study, we identified chlorogenic acid as one of the tyrosinase inhibitory (anti-whitening related) efficacy factors. Content of chlorogenic acid and the extent of tyrosinase inhibition were the highest in 80% *C. tricuspidata* fruit extract. In the previous report, chlorogenic acid was reported to affect melanogenesis through tyrosinase inhibition when converted into metabolites in cells [11]. Therefore, we established chlorogenic acid asthe biomarker of *C. tricuspidata* fruit. HPLC chromatograms revealed chlorogenic acid asthe major component of *C. tricuspidata* fruit.

In the present study, total phenolic and total flavonoids content of *C. tricuspidata* fruit extracts were compared. Total phenolic and flavonoids were the highest in 100% ethanolic extracts. Extracts containing phenolic compounds have been reported to exhibit tyrosinase inhibition [15,16,17].

Tyrosinase inhibition was the highest in 80% ethanolic extract. Besides, total phenolic and total flavonoid levels were highest in 100% ethanolic extract, thus indicating that the 80% ethanolic extract contains unknown tyrosinase inhibitors. It is hypothesized that through further studies, unknown tyrosinase inhibitors in 80% ethanolic extracts can be identified (Table 5).

## 3. Experimental Section

### 3.1. Plant Material and Preparation of the Extract

*C. tricuspidata* fruit was collected in May 2017 near Naju, Jeonnam Province, Korea. A voucher specimen (MNUCSS-CTF-01) was deposited in the College of Pharmacy, Mokpo National University. Fruits were dried and used for extract preparation. The air-dried and powdered *C. tricuspidata* fruits (10 g) were subjected to extraction twice with 20–100% ethanol (100 mL) at room temperature for three days. The 0% extract was prepared using hot water extraction (100 °C, 4 h). After filtration, the resultant ethanol solution was evaporated, freeze-dried, and stored at ‒50 °C. The crude extract was resuspended in ethanol and filtered using a 0.4 μm membrane. All samples were used for the optimization of the extraction process and in vitro experiments.

### 3.2. Instrumentation and Chromatographic Conditions

The HPLC conditions were established as shown in Table 6.

### 3.3. Preparation of Standards and Sample Solutions

Accurately weighed appropriate amounts of chlorogenic acid was mixed and dissolved in methanol in a 50 mL volumetric flask, to obtain a stock solution of 50 μg/mL. Solutions were subsequently 2-fold serially diluted to 3.125 μg/mL.

Samples (0.5 g) were dissolved in methanol (10 mL). Subsequently, 1 mL was diluted with 9 mL of mobile phase A to obtain a final solution with a known concentration of 25 mg/mL [18].

### 3.4. Method Validation

The analytical method usedfor the quantification of chlorogenic acid in the various extract of *C. tricuspidata* fruit was validated in terms of specificity, linearity, sensitivity, accuracy, precision, and recovery. Experiments were performed as previously described [16].

### 3.5. Analysis of the Extract from C. tricuspidata Fruit

The HPLC method developed herein was used to quantitatively determinate of the amounts of chlorogenic acid in 6 extracts from *C. tricuspidata* fruit.

### 3.6. Cell Viability

Human melanoma cells (B16F10) were purchased from the American Tissue Culture Collection (Manassas, VA, USA). Cells were seeded in 96-well plates and treated with 100 μg/mL of sample for 24 h. MTT was added and plates were incubated at 37 °C for 2 h. After dissolving the formazan crystals in 100 μL of DMSO, the absorbance was measured at 490 nm using an Enspire Multimode Plate reader (Perkin-Elmer, Akron, OH) [19].

### 3.7. Tyrosinase Inhibitory Assay

Tyrosinase inhibition assay was performed following a previously described method with some modification [20]. Briefly, reaction mixtures (total volume of 150 μL) with 49.5 μL of phosphate buffer (pH 6.8, 100 mM), 45 μL of distilled water, and 5 μL of sample dissolved in DMSO (100 μg/mL) were prepared. This was followed by the addition of 0.5 μL of mushroom tyrosinase (10 units) and 50 μL of the substrate, mixed well, and incubated for 10 min at 37 °C. Absorbance was measured at 475nm.

The percent inhibition of the enzyme reaction was calculated as follows:Inhibition rate (%) = (B − S)/B × 100(1)
where B and S are the absorbance values for the blank and sample, respectively.

### 3.8. Determination of Total Phenolic Content

The total phenolic content was determined using Folin-Ciocalteu assay [18]. A 1 mL of sample solution (5 mg/mL) was mixed with 1 mL of 2% (*w*/*v*) Na_2_CO_3_ solution and 1 mL of 10% Folin-Ciocalteu phenol reagent. After 10 min, the absorbance was measured at 750 nm using a microplate reader (Perkin Elmer, Waltham, MA, USA). The phenolic content was calculated from calibration curve of gallic acid. The results were expressed as mg of gallic acid equivalents per g of sample.

### 3.9. Determination of Total Flavonoids

The total flavonoid content was determined based ona previously reported colorimetric method [18]. Briefly, a 0.5 mL aliquot of the sample solution was mixed with distilled water (2 mL) and subsequently with 5% NaNO_2_ solution (0.15 mL). After incubation for 5 min, a 0.15 mL aliquot of 10% AlCl_3_ solution was added to the mixture and after 5 min, 4% NaOH solution (2 mL) was added to the mixture. Water was added to the sample to bring the final volume to 5 mL, and the mixture was thoroughly mixed and allowed to stand for 15 min. The absorbance of the resultant mixture was measured at 415 nm. Then, the total flavonoid content was calculated as quercetin equivalents (mg quercetin/g extract) by reference to a standard curve (*r*^2^ = 0.999).

## 4. Conclusions

In our preliminary study, we identified chlorogenic acid in the fruits of *C. tricuspidata* as antioxidant compound using bioassay-guided purification. The validated HPLC method was developed and applied to confirm the presence of chlorogenic acid in *C. tricuspidata* fruit extracts. Various ethanolic extracts of *C. tricuspidata* fruit were prepared, and 80% ethanolic extract was found to exhibit the highest tyrosinase inhibitory activity. Besides, 100% ethanolic extract possessed the highest content of total phenolics and flavonoids. Based on the results, it is evident that the 80% ethanolic extract contains unknown tyrosinase inhibitors. Further studies are necessitated to identify the unknown tyrosinase inhibitors in 80% ethanolic extracts. The results suggest that the observed antioxidant and tyrosinase inhibitory activity of *C. tricuspidata* fruit extract could partially be attributed to the presence of marker compounds in the extract.We report ananalytical method for standardization and extraction optimization of *C. tricuspidata* fruit for the first time. Further investigations are needed to confirm the biological effect of *C. tricuspidata* fruit extract for the potential development of the extract as a cosmetic source.

## Figures and Tables

**Figure 1 molecules-24-03266-f001:**
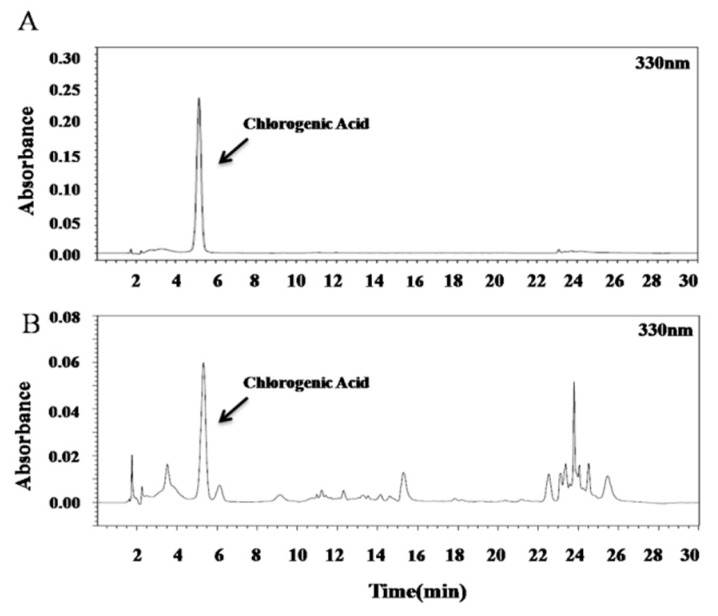
Analysis of *C. tricuspidata* fruit extracts by High Performance Liquid Chromatography (HPLC) method. (**A**) standard; (**B**) sample extract (fruit).

**Figure 2 molecules-24-03266-f002:**
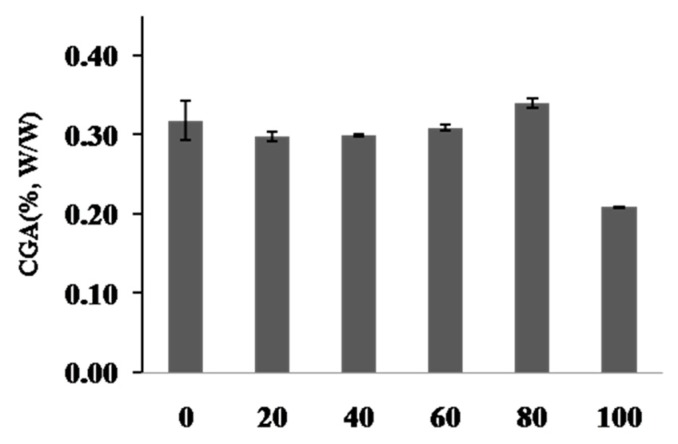
Content of chlorogenic acid (CGA) in hot water and ethanolic extracts from *C. tricuspidata* fruit. 0 (hot water ex); 20 (20% ethantol ex); 40 (40% ethanol ex); 60 (60% ethanol ex); 80 (80% ethanol ex); 100 (100% ethanol ex). Each value was the mean ± SD (*n* = 3).

**Figure 3 molecules-24-03266-f003:**
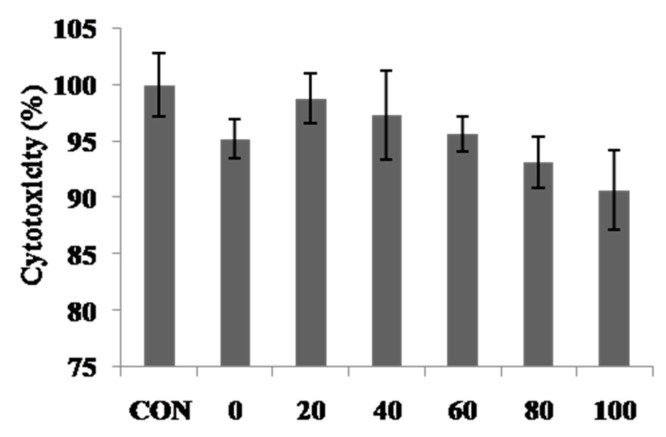
Cell viability of water and ethanolic extracts from *C. tricuspidata* fruit. 0–100 represent 0–100% ethanolic extract (100 μg/mL). Each value was the mean ± SD (*n* = 3).

**Figure 4 molecules-24-03266-f004:**
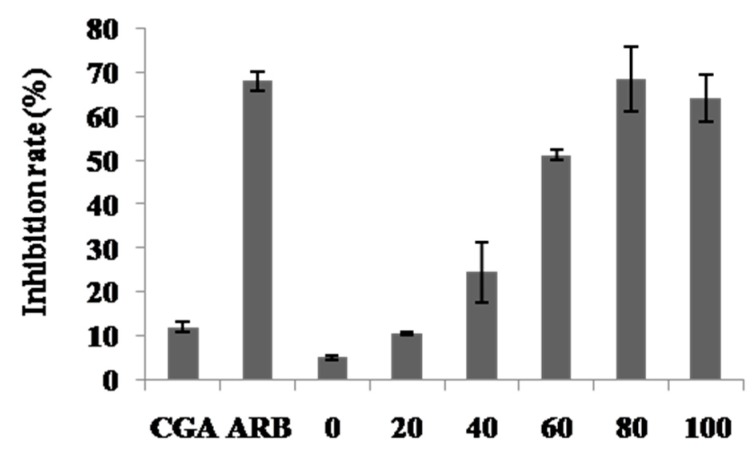
Tyrosinate inhibition in hot water and ethanolic extracts from *C. tricuspidata* fruit.CGA: chlorogenic acid (8 μg/mL); ARB (150 μg/mL): arbutin. 0–100 represent 0–100% ethanolic extract (100 μg/mL). Each value was the mean ± SD (*n* = 3).

**Table 1 molecules-24-03266-t001:** Chemical constituents and biological activities of *C. tricuspidata* fruit reported in previous literatures.

Constituent	Activity	Contents	Effective Dose (mg/kg/day)(route/animal)	Ref.
6,8-diprenylgenistein	Anti-obesity	Single compound	30 (oral/mouse)	[7]
Cudraisoflavones etc	Neuroprotective	*N.D	N.D	[2]
Genistein etc	Lipase inhibition	N.D	N.D	[5]
5,7,3′,4′-Tetrahydroxy-6,8-diprenylisoflavone	Antiallergy	N.D	N.D	[11]
Water extract	Dermatitis	Rutin was identified	60 (oral/mouse)	[3]
Gancaonin A etc	Monoamine oxidase inhibition	N.D	N.D	[6]
Water and etnaolic extract	Tyrosinase inhibition	Chlorogenic acid	N.D	This study
Scandenolone	Anti-cancer	Single compound	5 and 7.5 (intravenous/mouse)	[12]

*ND; not decribed.

**Table 2 molecules-24-03266-t002:** HPLC data for the calibration graphs and limit of quantification of the active compound.

Analyte	Retention Time (min)	*R* ^2^	Linear Range (μg/mL)	LOQ (μg/mL)	LOD (μg/mL)
Chlorogenic acid	4.7	0.998	3.125–50	2.11	0.7

**Table 3 molecules-24-03266-t003:** Analytical results of intra-day and inter-day precision and accuracy.

Analyte	Conc (μg/mL)	Intra-Day (*n* = 3)		Inter-Day (*n* = 3)	
		RSD (%) ^a^	Accuracy (%)	RSD (%)	Accuracy (%)
**Chlorogenic acid**	6.25 12.5 25	2.65 7.86 2.59	105.29 105.32 104.50	2.74 5.94 3.17	104.04 107.78 105.91

^a^ RSD: relative standard deviation.

**Table 4 molecules-24-03266-t004:** Analytical data of recovery (*n* = 6).

Analyte	Added (μg/mL)	Recovery (%) (Mean ± SD)	RSD (%) ^a^
Cholorogenic acid	6.25	103.39 ± 0.35	0.4
	12.5	96.79 ± 0.96	1.08
	25	98.59 ± 1.20	1.26

^a^ RSD: relative standard deviation.

**Table 5 molecules-24-03266-t005:** Antioxidant activity and total phenolic contents of *C. tricuspidata* fruit extracts.

Extract	Total Flavonoid (Ascorbic Acid eq. μg/100 μg Extract)	Total Phenolic Content (Gallic Acid eq. mg/g)
Hot water	7.9	31.9 ± 1.4
20% EtOH Ex	14.0	36.0 ± 3.0
40% EtOH Ex	10.8	29.9 ± 1.8
60% EtOH Ex	11.2	33.9 ± 2.1
80% EtOH Ex	19.5	35.9 ± 2.2
100% EtOH Ex	26.0	40.6 ± 2.7

**Table 6 molecules-24-03266-t006:** Analytical HPLC conditions for *C. tricuspidata* fruit extracts.

Parameters	Conditions		
Instruments and Column	Alliance 2695 HPLC system (Waters, Millford, MA, USA) Zorbax extended-C18 (C18, 4.6 mm × 150 mm, 5 µm)		
Flow rate	0.8 mL/min		
Injection volumn	10 μL		
UV detection	330 nm		
Run time	30 min		
**Gradient**	**Time (min)**	**A (%)**	**B (%)**
	0	10	90
	7	10	90
	8	20	80
	20	25	75
	21	100	0
	25	10	90
	30	10	90

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
