# Peer review of "Tyrosinase Inhibition Antioxidant Effect and Cytotoxicity Studies of the Extracts of Cudrania tricuspidata Fruit Standardized in Chlorogenic Acid"

_molecules, 2019, doi:10.3390/molecules24183266_

Round 1
Reviewer 1 Report
Chlorogenic acid is isolated from the leaves and fruits of dicotyledonous plants ¿because choosing a very common compound in plants as a biomarker? There may be another compound, specific to Cudrania tricuspidata fruit
Expand the introduction with the following publications on the subject
Lan-Ting Xin, Shi-Jun Yue, Ya-Chu Fan, Jing-Shuai Wu, Dan Yan, Hua-Shi Guan, Chang-Yun Wang (2017). Cudrania tricuspidata: an updated review on ethnomedicine, phytochemistry and pharmacology. RSC Adv., 7, 31807
Byung Woong Lee, Jin Hwan Lee, Sung-Tae Lee, Suh Awanchiri, Woo Song Lee, Tae-Sook Jeong, Ki Hun Park (2005). Antioxidant and cytotoxic activities of Xanthones from Cudrania tricuspidata. Bioorganic & Medicinal Chemistry Letters 15: 5548-5552
Jeong, CH., Choi, G.N., Kim, J.H. Kwak JH, Jeong HR, Kim DO, Heo HJ (2010). Protective effects of aqueous extract from Cudrania tricuspidata on oxidative stress-induced neurotoxicity. Food Sci Biotechnol 19: 1113.

Author Response
Response to the Reviewer #1’s Comments
(molecules-588614)
Chlorogenic acid is isolated from the leaves and fruits of dicotyledonous plants because choosing a very common compound in plants as a biomarker There may be another compound, specific to Cudrania tricuspidata fruitExpand the introduction with the following publications on the subject
Authors’ Response: We deeply appreciate the reviewer’s positive evaluation and constructive comments on our manuscript. We thank the reviewer for introducing a reference pertinent our work, the sentence has been added in the instruction with additive references (line 38-39). the revised texts in the manuscript are highlighted with red color.
Minor correction
line 2 anti-oxidnat-> antioxidant
line 36 Cudramia tricuspidata -> C. tricuspidata
line 60 anti-oxidnat-> antioxidant
line87 anti-cancer-> anticancer
line138 ofC. tricuspidata-> of C. tricuspidata
line 157 asthe -> as the
line 175-176 100°C -> 100 °C
‒50°C -> ‒50 °C
line 219 2 hours -> 2h
line 214 1mL -> 1 mL
line230 C. tricuspidataas -> C. tricuspidata as
line232 ofchlorogenic acid in C. tricuspidatafruit -> of chlorogenic acid in C. tricuspidata fruit
line234 possessedthe -> possessed the

Reviewer 2 Report
Comments:
Please add different chromatograms from different real samples. There many compounds contain similar structures with Chlorogenic Acid. How to confirm the selectivity and specificity of the proposed method? Authors only used the proposed method to detect one analyte, the run time 30 min is too long. The LC conditions could be modified.
Author Response
Response to the Reviewer #2’s Comments
(molecules-588614)
Please add different chromatograms from different real samples. There many compounds contain similar structures with Chlorogenic Acid. How to confirm the selectivity and specificity of the proposed method? Authors only used the proposed method to detect one analyte, the run time 30 min is too long. The LC conditions could be modified.
Authors’ Response: We thank the reviewer for pointing this out. According to reviewers suggestion, we tried to add all the chromatograms of the assay samples to Figure 1, but the chromatogram was too large and complex.The reviewer pointed the analysis time (30min). In fact, we applied sufficient analysis time to identify rutin and 6,8-Diprenylgenistein reported in the references (including chlorogenic acid). However, we could not identified rutin and 6,8-Diprenylgenistein in the extract and described in the discussion section (line94-111). We ask the reviewer’s generous understanding on this matter.
